# An Exploratory Assessment of Factors with Which Influenza Vaccine Uptake Is Associated in Hungarian Adults 65 Years Old and Older: Findings from European Health Interview Surveys

**DOI:** 10.3390/ijerph19127545

**Published:** 2022-06-20

**Authors:** Gergő József Szőllősi, Nguyen Chau Minh, Cornelia Melinda Adi Santoso, Judit Zsuga, Attila Csaba Nagy, László Kardos

**Affiliations:** 1Department of Interventional Epidemiology, Faculty of Public Health, University of Debrecen, 4032 Debrecen, Hungary; nguyen.chauminh@sph.unideb.hu (N.C.M.); cornelia.melinda@sph.unideb.hu (C.M.A.S.); nagy.attila@sph.unideb.hu (A.C.N.); 2Department of Habilitational Medicine, Faculty of Public Health, University of Debrecen, 4032 Debrecen, Hungary; zsuga.judit@med.unideb.hu; 3Department of Biostatistics and Bioinformatics, Faculty of Public Health, University of Debrecen, 4032 Debrecen, Hungary; l.kardos@orvosbiostat.hu

**Keywords:** immunization, vaccination coverage, influenza, Hungary, elderly population

## Abstract

Influenza vaccination is an imperative public health task for elderly people due to a higher risk of developing more severe complications. The main aim of our study was to determine the influencing factors of being vaccinated against influenza among subjects aged 65 and above. Data were from the Hungarian implementations of the European Health Interview Survey 2009, 2014 and 2019 studies with a final sample size of 3355. A multivariate logistic regression model with interactions was used to identify the possible factors associated with vaccination. Approximately 32% of the participants were vaccinated for the most recent influenza season. The most important factors were identified that contributed to influenza vaccination among individuals, which were the following: educational attainment, having a partner, the annual frequency of specialist and doctor visits, and having comorbidities. Respondents who thought that they could do a lot for their health had higher odds of being immunized. Being obese seemed to be a risk factor. According to our findings, the current influenza vaccination coverage was considered as low in Hungary; hence, the implementation of minor reformulations in the field of health policy is suggested.

## 1. Introduction

According to the World Health Organization (WHO), seasonal influenza causes an estimated 3–5 million cases of severe illness worldwide and approximately 290,000–650,000 respiratory deaths annually [1]. Approximately 5–10% of adults and 20–30% of children are infected each year by influenza. The infection increases the risks of hospitalization, severe exacerbations, and death. Immunization has several benefits, including protection against non-communicable diseases (heart attack, stroke, etc.) and their complications [2]. Vaccine-preventable illnesses—such as influenza—are more likely to cause serious consequences in older people. Therefore, influenza is one of the most critical global public health challenges; it is regarded as a major contributor to and one of the leading causes of mortality and morbidity, particularly among older people [3,4,5,6].

Elderly people are more susceptible to the fatal outcome of influenza. In this demographic, mortality and morbidity both rise during seasonal influenza epidemics, making them a high-priority target for vaccines even though vaccine efficacy may vary from year to year, and does not encompass the prevention of infections [6]. However, the overall effectiveness of influenza vaccination could be influenced by how the flu virus strains in the vaccine match the predominantly circulating strains. Furthermore, several studies have shown the efficacy of influenza vaccination in reducing death and other adverse outcomes [7,8,9,10]. Influenza vaccination is considered to be a cost-saving method in cost-effectiveness ratios in high- and upper-middle-income countries [11]. To mitigate the disease burden of influenza at the level of primary prevention, vaccination is highly advisable. The WHO and the Centers for Disease Control and Prevention (CDC) recommend annual influenza vaccination for all persons aged ≥6 months [6,12], especially for people aged 65 or above. This recommendation is endorsed by Hungarian national public health authorities [13]. The elderly, people with chronic conditions and other people at high risk can receive free-of-charge influenza vaccination in almost all European Union countries, including Hungary. Despite the fact that vaccination is arguably the most effective method for preventing infection [14], vaccination coverage is insufficient and unsatisfactory among patients aged 65 years or older, as suggested by data obtained from the Organisation for Economic Co-operation and Development (OECD) [15]. In order to achieve the greatest benefit from influenza vaccination, uptake in high-risk groups should approach 75% [16,17]. However, remarkable variability (2–85%) has been observed in global influenza vaccination coverage, according to the OECD [15] and the European Centre for Disease Prevention (ECDC) [18]. The situation is further aggravated by the decline in influenza vaccine uptake among people aged 65 years or older [15], despite various guidelines and protocols continuing to recommend influenza immunization. 

The potential reason for low influenza vaccination coverage could be the lack of standardized protocols for vaccine distribution. Another possible explanation lies in the public’s and health professionals’ diverse attitudes towards vaccination, including skepticism regarding its efficacy, which represents a target area for health education programs [19,20]. Although several studies have investigated the factors influencing vaccination in general, the actual reasons for low vaccine uptake may vary widely across countries [21], aggravating the challenge posed by the lack of global monitoring indicators for vaccine uptake among high-risk individuals. Increasing vaccination coverage by any reasonable means is an imperative interdisciplinary public health task [22]; therefore, it is essential to understand which factors influence vaccine acceptance in each specific target population. 

### Aims

The main objective of this study was to identify the possible factors that influence influenza vaccination by analysing questionnaire responses provided by a nationwide representative sample in Hungary. The secondary objective was to describe the patterns of influenza vaccination coverage in Hungary based on European Health Interview Survey (EHIS) studies, with a special focus on residents aged 65 years and older.

## 2. Materials and Methods

### 2.1. Database

Our data were collected from the 2009, 2014 and 2019 implementations of the cross-sectional European Health Interview Survey in Hungary, which were carried out on representative samples using a standardized questionnaire supervised by Eurostat [23,24]. The databases are not available to the public but can be requested from the institution that performed and supervised the data collection and primary analysis, i.e., the Central Statistical Office of Hungary [25]. EHIS was based on stratified two-step probability samples selected to establish reliable estimations related to health status indicators of the Hungarian population aged 15 years and older from private households. Because the methodology executed during primary data collection was unchanged, the three datasets and their identical sets of variables were considered comparable. All databases are representative of the Hungarian population.

### 2.2. Data

The main outcome of our study was based on the following question: ‘Have you been vaccinated against influenza within a year?’. Based on the answers, we identified two groups. The first group consisted of respondents who had received a flu shot within the last year, and the second consisted of those who had received their last influenza vaccination more than 1 year earlier or had never received the influenza vaccine. As the effectiveness of the vaccine varies from year to year (which is why the vaccine should be administered annually), we considered those who have never been vaccinated and those who have not been vaccinated against influenza in the given season sufficiently similar in protection status for pooling together. The database contained information about the respondents’ socio-demographic characteristics, including gender (male or female), highest educational level (primary, secondary or tertiary) and marital status (has support or has no support). To adjust our analysis for potential confounders originating from territorial heterogeneity, we used a variable for the geographic region of residence (Central Hungary, Southern Great Plain, Southern Transdanubia, Central Transdanubia, Western Transdanubia, Northern Great Plain or Northern Hungary). Self-perceived health status responses (‘good’ or ‘bad’) were also used in the analysis. We used responses to the question ‘How much can you do for your health?’ (‘not so much’ or ‘a lot’) to assess the effect of health-related attitudes and optimism. Lifestyle factors were also included in the analysis, such as smoking status (current and occasional smoker vs. non-smoker) and body mass index (not overweight (BMI < 25) vs. overweight and obese (BMI ≥ 25)) based on self-reported body weight and height. Satisfaction related to healthcare services was represented by responses to ‘How satisfied are you with your specialist?’ and ‘How satisfied are you with your doctor?’ (‘satisfied’ or ‘not satisfied’). Healthcare-related attitude and compliance were included in the analysis through responses to ‘When was your last meeting with your specialist?’ and “When was your last meeting with your doctor?’ (‘within a year’ or ‘more than a year ago’). Self-reported diseases were pooled into three major groups: asthma and chronic obstructive pulmonary disease (COPD), bronchitis and emphysema were collectively considered ‘respiratory diseases’; myocardial infarction, coronary heart disease, hypertension, stroke, atrial fibrillation and other heart diseases were ‘cerebro- and cardiovascular diseases’; and diabetes and lipid metabolism disorders were ‘endocrine diseases’. The effects of indicator variables (‘yes’ or ‘no’) for the presence of comorbidity (any ongoing constituent disease vs. no disease) in each major group were analysed. We also used an indicator for the calendar year of primary data collection (2009, 2014, or 2019).

### 2.3. Statistical Methods

Pearson’s chi-square test was performed on the merged database to assess the unadjusted differences between vaccinated and non-vaccinated persons in terms of categorical characteristics. A multivariate logistic regression model was used to identify the factors that might influence vaccination status against influenza among respondents aged 65 years or older. We checked for interactions between explanatory variables and found 12 potential significant pairs. These were then checked in combination, and three were taken into consideration when the multivariate model was established; therefore, the model contained three interaction terms (between ‘Year’ and ‘When did you last meet with your doctor?’; ‘Education’ and ‘Self-perceived health status’; ‘Marital status’ and ‘When did you last meet with your doctor?’). The survey year variable was also included to assess changes in vaccine uptake over time. The results are expressed as adjusted odds ratios (OR) and *p*-values. The goodness of model fit was examined using the Hosmer–Lemeshow test. Statistical analysis was performed using Stata Statistical Software (version 9.0, Stata Corp, College Station, TX, USA), and *p* < 0.05 was considered significant.

## 3. Results

The original sample sizes were 5051 (2009), 5826 (2014) and 5603 (2019) for the full datasets covering all age groups. In 2009, there were 1046 respondents aged 65 or older (21%); the figures for 2014 and 2019 were 1216 (21%) and 1628 (29%), respectively. After pooling the age-filtered datasets (*n* = 3890), we excluded those respondents (*n* = 535; 14%) who had not answered all questions relevant to our research as they would not have contributed to multivariate regression analysis; therefore, the final sample size was 3355. The sample consisted of 1269 (38%) males and 2086 (62%) females. The most frequent educational level was secondary, with 1807 (54%) respondents, which was followed by primary, with 1019 (30%) respondents. The tertiary educational level was the least frequent, with a prevalence of 16% (*n* = 529). Regarding marital status, 1656 (49%) respondents had social support, and 1699 had no social support. Self-perceived health status was considered ‘good’ by 2392 (71%) elderly people, and 963 (29%) reported that they considered their health ‘bad’. Approximately 65% (*n* = 2181) of the respondents said that they could do ‘a lot’ for their health. Normal or underweight body mass index (BMI) was observed in 998 (30%) cases, and 2357 (70%) people were considered overweight or obese. The prevalence of smoking was 40% in the sample; 1348 people were considered smokers during primary data collection. Satisfaction with specialist(s) was reported by 2524 (75%) respondents, and 2945 (88%) were satisfied with their primary care doctor. The most recent doctor visit was within the last year for 3082 (92%) elderly people, and 2461 (73%) met with their specialist within the last year. Respiratory diseases affected 449 (13%) people aged 65 years or older, cardio- and cerebrovascular diseases affected 2623 (78%), and endocrine diseases affected 1262 (38%). The number of respondents varied from region to region; the smallest proportion of the respondents was from the region of Southern Transdanubia (*n* = 322, 10%), and the greatest proportion was from Central Hungary (*n* = 938, 28%).

### 3.1. Vaccination Coverage of Residents Aged 65 Years or Older

The vaccination coverage rates of 3355 residents aged 65 years or older were: 37% (*n* = 352) in 2009, 32% (*n* = 348) in 2014 and 28% (*n* = 366) in 2019. Approximately 32% of the participants aged 65 years and older had been vaccinated and were considered to be protected for the most recent influenza season. A significant decreasing trend in vaccination was observed over the years (*p* < 0.001), indicating that vaccination coverage has recently decreased among people aged 65 or above. Men (*n* = 429, 34%) tended to have a significantly higher vaccination coverage compared with women (*n* = 637, 31%) (*p* = 0.049). The highest vaccination coverage was observed among respondents with a tertiary level of education (*n* = 211, 40%), which was followed by secondary (*n* = 554, 31%) and primary (*n* = 301, 30%) education (*p* < 0.001). People with social support (i.e., married) did not have significantly higher vaccination coverage (*n* = 563, 33%) compared to those who lived alone (*n* = 503, 30%) (*p* = 0.086). Respondents with bad self-perceived health status (*n* = 335, 35%) were more likely to be vaccinated compared with respondents with good self-perceived health status (*n* = 731, 31%) (*p* < 0.017). Those respondents who stated that they could do a lot for their health did not have significantly higher vaccination rates (*n* = 703, 32%) compared to those who answered that they could not do so much (*n* = 363, 31%) (*p* = 0.436). Vaccination rates were significantly higher in people with normal BMI (*n* = 344, 34%) than in obese (*n* = 722, 31%) (*p* = 0.029). Non-smokers tended to have higher proportions (*n* = 651, 32%) of vaccinated status compared to smokers (*n* = 415, 31%), but the association was not significant (*p* = 0.314). Respondents who were satisfied with their doctors (*n* = 955, 32%) had significantly higher vaccination coverage than those who were unsatisfied (*n* = 111, 27%) (*p* = 0.029). Satisfaction with specialists had no significant association with vaccination (*p* = 0.301); however, those who were satisfied achieved higher vaccinated rates compared with those who were not satisfied. Respondents who visited their doctor (*n* = 1028, 33%) or their specialist (*n* = 860, 35%) in the last year had significantly higher vaccination rates than those who did not visit their doctor or specialist (*p* < 0.001). Respondents who had accompanying respiratory (*n* = 173, 39%), cardio- or cerebrovascular (*n* = 886, 34%), or endocrine (*n* = 464, 37%) disease had significantly higher vaccination coverage compared with those who did not suffer from these comorbidities (*p* < 0.001). Territorial heterogeneity was also observed, with vaccination coverage ranging between 28 and 35%, but the differences were not significant among the respondents (*p* > 0.05). Vaccination coverage of respondents aged 65 years or older can be seen in Table 1.

### 3.2. Cross-Table Analysis of Vaccination Status

We calculated the vaccination status broken down by marital status and last appointment with the doctor. We found that people with no support who had not met with their doctor within a year had the lowest rate of vaccination (8%), in contrast with a vaccination rate of 33% in unpartnered people aged 65 or above who had visited their doctor within a year. The same association was observed among those who had a partner, of whom those who had visited their doctor within a year had higher vaccination frequency (34%) compared with those who did not meet with their doctor regularly (21%).

Comparing vaccination across levels of education and self-perceived health status, we found that a higher educational level was a protective factor in all cases. The cohort of respondents aged 65 or older who stated that they had a ‘bad’ health status had a higher (primary educated = 32%, secondary = 34% and tertiary = 61%) vaccination rate compared with those who stated that their health status was ‘good’ (primary = 28%, secondary = 30% and tertiary = 37%).

The year of the EHIS study was in interaction with the last appointment with the doctor; we observed that all the respondents had a higher rate of vaccinated status if they had met with their doctor within a year (2009 = 40%, 2014 = 34% and 2019 = 29%) compared with those who did not visit their doctor regularly (2009 = 11%, 2014 = 12% and 2019 = 19%).

### 3.3. Multivariate Model with Interactions

The goodness of fit test executed on the multivariate model consisted of data on the age group 65 years or older and was considered a good fit (*p* = 0.551).

Patients who had visited their doctor within the last year in 2019 had significantly lower odds of being immunized against influenza compared with patients who had visited their doctor within the last year in 2009 (OR = 0.56, *p* < 0.001). The participants who visited their doctor within the last year in 2009 had significantly higher odds of being vaccinated against influenza compared with those who did not visit their doctor within the last year (OR = 4.70, *p* < 0.001). The same association could be seen in 2014, where people aged 65 or above who had visited their doctor within the last year had 2.81 higher odds of being vaccinated against influenza compared with those who had not visited their doctor within the last year (*p* = 0.002). People who had visited their specialist within the last year had higher odds of being immunized against influenza compared with those who had not visited their specialist within the last year (OR = 1.48, *p* < 0.001). Influenza immunization showed a significant association with the level of education among people with bad self-perceived health status: compared to those with primary education, respondents with tertiary education had greater odds of being vaccinated (OR = 3.67, *p* < 0.001). This relationship was also observed in terms of good self-perceived health status, where respondents with a tertiary educational level had 46% higher odds of being vaccinated compared with those with a primary education level (OR = 1.46, *p* = 0.011). Participants with good self-reported health status who had secondary education had significantly lower odds of being immunized compared with people with bad self-perceived health status (OR = 0.40, *p* < 0.001). People aged 65 or older who had a partner and had visited their doctor in the last year had significantly higher odds of having received a flu shot compared with those who had no support (OR = 2.76, *p* = 0.010). The unpartnered respondents who had visited their doctor within the last year had 4.33 times higher odds of being immunized compared with those who did not visit their doctor in the last year (OR = 4.33). Being obese seemed to be a risk factor for not being vaccinated; the obese or overweight respondents had a significantly lower odds of being immunized against the virus within a year compared with people with normal BMI (OR = 0.76, *p* = 0.004). There was a significant relationship between being vaccinated and having comorbidity, including respiratory disease (OR = 1.29, *p* = 0.019), cardiovascular or cerebrovascular disease (OR = 1.35, *p* = 0.004), and endocrine (OR = 1.25, *p* = 0.005) disease; people aged 65 or above suffering from these medical conditions tended to have higher odds of being immunized against influenza. No significant relationship was found between vaccination and gender; however, it appeared that men had greater odds of being immunized than women (OR = 1.17, *p* = 0.077). A similar weak association was found between responses to the question “How much can you do for your health” and vaccination: those who said they could do a lot for their health had 19% greater odds of having had a flu shot (OR = 1.19, *p* = 0.056), but this relationship did not reach significance. Smoking status (*p* = 0.082) and satisfaction with the doctor (*p* = 0.065) showed no relationship with vaccination. Frequent doctor visits of partnered elderly people showed some signs of association with influenza immunization: elderly people with frequent doctor visits had greater odds of being immunized compared with those who did not visit their doctor regularly (OR = 1.56), but this association was not significant either (*p* = 0.056). Being satisfied with the specialists had no significant effect on vaccination (*p* = 0.843). Territorial heterogeneity was not observed regarding vaccination. The results of the multivariate model can be seen in Table 2.

## 4. Discussion

Even though the influenza vaccination is free of charge for residents aged 65 or above, and it is recommended by public health authorities, the vaccination coverage during the years studied was considered insufficient in Hungary. This study also found a significantly decreasing trend in vaccination coverage across the years, which was considerably lower than the European Union’s recommended rate of 75% [16]. Similarly low vaccination coverage was reported in 2016–2017 for older populations in other European countries based on the technical report of ECDC related to seasonal influenza vaccination coverage in EU/EEA member states [18]. This was further exacerbated by the observation that the influenza vaccination coverage rate was considered insufficient among other high-risk groups.

Based on our research, we identified the most essential factors that contributed to influenza vaccination among Hungarian residents aged 65 or older. A higher educational level was shown to be a protective factor, and this finding was corroborated by the literature. However, other studies found that a primary level of education was a protective factor compared with no schooling; in our study, we found this difference between tertiary and primary levels. Therefore, a higher level of education was shown to be related to vaccine uptake [26]. Marital status—especially having a partner—seemed to be a protective factor in terms of vaccination. The male gender had no significant association with vaccine uptake; other studies have reported differences in vaccination between genders [27,28,29]. People who thought that they could do a lot for their health had non-significantly higher odds of being immunized; a similar association was reported in a recent study, where an important factor was the past behaviour towards vaccination [27]. The annual frequency of specialist and doctor visits was a significant influencing factor for immunization, and the same relationship has been seen in the literature [30,31,32]. We highlight that regular visits with a doctor or specialist could indicate the presence of a more serious health condition that may encourage further follow-up of the recommendations by the care staff. Comorbidities, such as respiratory, cardio- or cerebrovascular, and endocrine diseases, in elderly persons had a significant association with influenza vaccination, which corresponds with the international recommendations, as these high-risk groups should be immunized against the influenza virus [12,33]. Despite the fact that obese people are likely to develop more severe influenza infection compared with normal-weight people, they tended to have a lower willingness for vaccine uptake in our study; however, the opposite association was observed in a recent study [34]. Regional heterogeneity in vaccination status was observed in the analysis; the vaccination coverage varied between the counties, but this variation had no significant effect on the willingness for vaccine uptake.

### Strengths and Limitations

Although EHIS did not primarily focus on identifying the reasons for low vaccine coverage, the strongest candidate determinants of influenza vaccination were exploratively identified by our supplementary analysis of this valuable body of data. A relatively complex multivariate logistic regression model was run on a merged dataset that consisted of three representative samples to determine the possible factors that could correlate with being properly vaccinated against the influenza virus; however, there were some self-perception-based questions (e.g., the registered comorbidities, BMI), which could have resulted in under-representation in our results. Therefore, the reliance on socio-demographic and satisfaction questions should be noted. Furthermore, mild and severe diseases were not distinguished from each other due to a lack of data. Due to the nature of the data collection, the available database only contained information on responders; in the absence of non-responder data, potential systematic differences between responders and non-responders were not possible to assess.

## 5. Conclusions

The risks associated with not having proper immunization were identified in this study. Primary care should focus on obese patients who do not have comorbid chronic diseases, who have primary education and who do not see their doctor regularly. Vaccination should be considered a priority for public health agencies and primary care units, and it appears that the provided care and its quality are not properly developed in this field in Hungary. The situation is further aggravated by the fact that there are no special vaccination centres in Hungary, even though these could arguably help increase vaccine coverage; instead, vaccination is primarily done at the level of primary care (this approach is fundamentally different than the country-wide immunization effort response to the COVID-19 situation was where vaccination centres were set up in hospitals with intense supportive involvement of the military in logistics, organization and security related functions—a potential source of lessons to be learned when it comes to vaccination against other diseases). As a result, the healthcare professionals in Hungary are unable to take advantage of possible avenues to enhance influenza vaccination coverage. Because of this, immediate action is needed in this field to improve the quality of organized care. Influenza vaccination should be recommended to reduce the onset of severe complications and the risk of death.

## Figures and Tables

**Table 1 ijerph-19-07545-t001:** Descriptive statistics by vaccination status of survey respondents aged 65 or older.

Variable	Level	Vaccination Status (*n*, %)		*p*-Value
		Unvaccinated	Vaccinated	
Year	2009	597 (63%)	352 (37%)	<0.001
	2014	744 (68%)	348 (32%)	
	2019	948 (72%)	366 (28%)	
Gender	Female	1449 (69%)	637 (31%)	0.049
	Male	840 (66%)	429 (34%)	
Education level	Primary	718 (70%)	301 (30%)	<0.001
	Secondary	1253 (69%)	554 (31%)	
	Tertiary	318 (60%)	211 (40%)	
Marital status	Has no support	1153 (70%)	503 (30%)	0.086
	Has support	1136 (67%)	563 (33%)	
Self-perceived health status	Bad	628 (65%)	335 (35%)	0.017
	Good	1661 (69%)	731 (31%)	
How much can you do for your health?	Not so much	811 (69%)	363 (31%)	0.436
	A lot	1478 (68%)	703 (32%)	
Body mass index	Not overweight (BMI < 25)	654 (66%)	344 (34%)	0.029
	Overweight or obese (BMI ≥ 25)	1635 (69%)	722 (31%)	
Smoking status	Non-smoker	1356 (68%)	651 (32%)	0.314
	Smoker	933 (69%)	415 (31%)	
How satisfied are you with your specialist?	Satisfied	1710 (68%)	814 (32%)	0.301
	Not satisfied	579 (70%)	252 (30%)	
How satisfied are you with your doctor?	Satisfied	1990 (68%)	955 (32%)	0.029
	Not satisfied	299 (73%)	111 (27%)	
When did you last meet with your doctor?	More than a year ago	235 (86%)	38 (14%)	<0.001
	Within a year	2054 (67%)	1028 (33%)	
When did you last meet with your specialist?	More than a year ago	688 (77%)	206 (23%)	<0.001
	Within a year	1601 (65%)	860 (35%)	
Region	Central Hungary	635 (68%)	303 (32%)	0.471
	Southern Great Plain	316 (71%)	130 (29%)	
	Southern Transdanubia	213 (66%)	109 (34%)	
	Central Transdanubia	246 (68%)	118 (32%)	
	Western Transdanubia	246 (72%)	98 (28%)	
	Northern Great Plain	357 (69%)	162 (31%)	
	Northern Hungary	276 (65%)	146 (35%)	
Respiratory disease	No	2013 (69%)	893 (31%)	0.001
	Yes	276 (61%)	173 (39%)	
Cardiovascular or cerebrovascular disease	No	552 (75%)	180 (25%)	<0.001
	Yes	1737 (66%)	886 (34%)	
Endocrine disease	No	1491 (71%)	602 (29%)	<0.001
	Yes	798 (63%)	464 (37%)	

**Table 2 ijerph-19-07545-t002:** Factors that influenced influenza vaccination among the residents aged 65 or above in Hungary based on the multivariate logistic regression model.

Factor (Stratum, If Any)	Level	Adjusted Odds Ratio	*p*-Value
Year (subjects of infrequent doctor visits)	2009		
	2014	1.51	0.413
	2019	1.69	0.256
Year (subjects of frequent doctor visits)	2009		
	2014	0.90	0.569
	2019	0.56	<0.001
Doctor visits (year 2009)	Infrequent doctor visits		
	Frequent doctor visits	4.70	<0.001
Doctor visits (year 2014)	Infrequent doctor visits		
	Frequent doctor visits	2.81	0.002
Doctor visits (year 2019)	Infrequent doctor visits		
	Frequent doctor visits	1.56	0.144
Gender	Female		
	Male	1.17	0.077
Education level (good self-perceived health status)	Primary		
	Secondary	1.01	0.435
	Tertiary	1.46	0.011
Education level (bad self-perceived health status)	Primary		
	Secondary	1.19	0.264
	Tertiary	3.67	<0.001
Self-perceived health status (primary education level)	Bad		
	Good	0.99	0.948
Self-perceived health status (secondary education level)	Bad		
	Good	0.40	<0.001
Self-perceived health status (tertiary education level)	Bad		
	Good	0.92	0.482
Marital status (subjects of frequent doctor visits)	Has no support		
	Has support	0.99	0.940
Marital status (subjects of infrequent doctor visits)	Has no support		
	Has support	2.76	0.010
Frequency of doctor visits (has marital support)	Infrequent doctor visits		
	Frequent doctor visits	1.56	0.056
Frequency of doctor visits (has no marital support)	Infrequent doctor visits		
	Frequent doctor visits	4.33	<0.001
How much can you do for your health?	Not so much		
	A lot	1.19	0.056
Body mass index	Not overweight (BMI < 25)		
	Overweight or obese (BMI ≥ 25)	0.78	0.004
Smoking status	Non-smoker		
	Smoker	0.76	0.082
How satisfied are you with your specialist?	Satisfied		
	Not satisfied	1.02	0.843
How satisfied are you with your doctor?	Satisfied		
	Not satisfied	0.79	0.065
When did you last meet with your specialist?	More than a year ago		
	Within a year	1.48	<0.001
Region	Central Hungary		
	Southern Great Plain	0.95	0.716
	Southern Transdanubia	1.20	0.201
	Central Transdanubia	1.20	0.411
	Western Transdanubia	0.92	0.566
	Northern Great Plain	1.09	0.513
	Northern Hungary	1.29	0.055
Respiratory disease	No		
	Yes	1.29	0.019
Cardiovascular or cerebrovascular disease	No		
	Yes	1.35	0.004
Endocrine disease	No		
	Yes	1.25	0.005

## Data Availability

The data presented in this study are not available to the public but can be requested from the institution that performed and supervised the data collection and primary analysis: Hungary’s Central Statistical Office.

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
