# Peer review of "An Exploratory Assessment of Factors with Which Influenza Vaccine Uptake Is Associated in Hungarian Adults 65 Years Old and Older: Findings from European Health Interview Surveys"

_ijerph, 2022, doi:10.3390/ijerph19127545_

Round 1
Reviewer 1 Report
Dear Authors,
Thank you very much for your extensive changes and modifications. The MS now looks much better than the last version. Re revision is satisfactory.
Author Response
Dear Reviewer, we would like to thank you for giving us the opportunity to submit a revised draft of our manuscript.
Reviewer 2 Report
Thank you for improving the manuscript. I have not any additional concerns.
Author Response

(The authors gave the same response as above.)

Reviewer 3 Report
Major comments
1. The table matrix is extraordinarily complex and hard to understand.
When I added the prevalence of each variable in Table 2-4, the total percentage did not become 100%.
2. If the author wants to show associated factors between with and without influenza vaccination in those aged 65 or above, they should show only Tables 1 and 5. Please focus on the aim of the study.
3. The authors performed a multiple regression analysis in Table 5
However, there is numeral multicollinearity (e.g., year, educational level, and mental status). In addition, the definition of the factors is obscure (e.g., BMI, smoking status, diseases)
Author Response
Dear Reviewer, we are grateful for the comments and would like to thank you for giving us the opportunity to submit a revised draft of our manuscript.
We agree with the Reviewer that Tables 2-4 unintendedly seem complex and add little to the understanding of the results. We removed these tables from the manuscript, as advised. As a side note, the percentage figures of these tables were never expected to add up to 100% since they were stratum-specific vaccination levels, not subdivisions of some total quantity across strata. We also made minor modifications in section 3.2. to ensure that the removed tables’ intended message is fully conveyed through text.
We appreciate the Reviewer’s insightful suggestion regarding multicollinearity; indeed, collinearity is a caveat as it can reduce the precision of the estimation of coefficients, but we believe that it could be an issue if the regression had failed to produce significant relationships. The case is quite the opposite.
We kindly apologize if the definition of several factors seemed to be obscure; we have modified the text as advised to make things clearer. For this reason, we decided to make the following changes in the manuscript:
‘Lifestyle factors were also included in the analysis, such as smoking status (current and occasional smoker vs non-smoker) and Body Mass Index (not overweight (BMI<25) vs overweight and obese (BMI≥25)) based on self-reported body weight and height.’
‘Self-reported diseases were pooled into three major groups: asthma and chronic obstructive pulmonary disease (COPD), bronchitis, and emphysema were collectively considered ‘respiratory diseases’; myocardial infarction, coronary heart disease, hypertension, stroke, atrial fibrillation and other heart diseases were ‘cerebro- and cardiovascular diseases’; and diabetes and lipid metabolism disorders were ‘endocrine diseases’.’
Round 2
Reviewer 3 Report
Minor comment
1. Please clarify the number of BMII in Table 1 (e.g., BMI≥25)
Author Response
Dear Reviewer,
Corrections have been made.
This manuscript is a resubmission of an earlier submission. The following is a list of the peer review reports and author responses from that submission.
Round 1
Reviewer 1 Report
Dear Authors, the work is impressive. But it must be improved.
Here are my suggestions-
- Is there any link or database or government regulations with those data sets! that's very crucial to believe the data sets.
- Table 1 should contain the age data.
- The discussion part must be modified with the novelty of the work, and its importance.
- Any motives for not being vaccinated or getting vaccinated must be declared.
- Ethical permission should be mentioned according to the GDPR rules of the EU.
Reviewer 2 Report
I would like the authors to address some concerns I have about the research:
Line 85: I wonder if the authors meant '65 years' instead of '15 years'.
Line 131: Please, mention which were the p-value thresholds for statistical significance and borderline significance.
Lines 255-256: Depending on the p-values thresholds, this sentence could or not be true.
Lines 255-260: If the threshold for statistical significance is 0.05, I think calling p-values of 0.077, 0.082 or 0.65 as 'borderline significant' instead of 'non-significant' could be somehow misleading.
Reviewer 3 Report
Influenza vaccination is an imperative public health task for elderly people due to a higher risk of developing more severe complications. The main aim of this study was to determine the influencing factors of being vaccinated against influenza among subjects aged 65 and above.
The manuscript is well written, has 5 tables and 30 references.
The authors are followed the guidelines of the manuscript structure.
The topic is interesting.
I think that the influenza vaccination should be performed in a special vaccination centrum instead of primary care, and the authors need to highlight this.
Moreover government regional offices and "healthcare organizers" should play bigger roles in organizing vaccination programs such as Covid-19 and influenza. COVID-19 situation proved, that vaccination centrums in the hospitals, organized and protected by the military are a perfect solution to make an easy, fast, professional, and structured vaccination process.
I believe that authors should reflect on these issues as well at least in the conclusion part.
I think that after that the manuscript is ready to be further processed.
Reviewer 4 Report
This study and manuscript involve an exploratory assessment of factors that are correlated with, and thus could influence, influenza vaccine acceptance among people 65 years old and older in Hungary. As noted in the Introduction, high uptake of influenza vaccine, particularly among older adults, can help prevent severe illness in people at higher risk for influenza complications. The data used in the study came from three cross-sectional European Health Interview Surveys (EHIS 2009, 2014, and 2019). Overall, approximately 32% of the 3,335 respondents had been vaccinated for the most recent influenza season, with the data indicating a statistically significant decline over time (from 37% in 2009 to 32% in 2014 to 28% in 2019).
This research and manuscript’s strengths include: 1) a focus on a significant public health issue; 2) well-stated study objectives, 3) a generally well-designed questionnaire, with measures assessing self-perceived health status, lifestyle factors, satisfaction with health care providers, self-reported chronic health conditions, and compliance, including with influenza vaccination recommendations; 4) the use of three EHIS datasets; and 5) Table 1. The manuscript is also well organized and generally well written.
Comments and suggestions for strengthening the manuscript are:
- I would highly recommend not characterizing people 65 years and older as “elderly.” Many people in this age group are healthy and active and respond negatively to being labelled “elderly.”
- The title would benefit from revision to something along the lines of “An Exploratory Assessment of Factors that Associated with Influenza Vaccine Uptake in Adults 65 Years Old and Older: Finding from Three National Surveys.” Note, since surveys were used it is not possible to identify cause-effect relationships, and as such, it is not possible to determine “influence.” It is, however, accurately stated in the manuscript that the analyses can identify factors that potentially influence influenza vaccination.
- It would help to be more accurate regarding the effectiveness of seasonal influenza vaccination, particularly among people 65 years old and older. First, influenza vaccine efficacy varies widely from year to year, and is often less than 40%. Second, effectiveness is based on how well vaccination protects people against severe influenza illness, such as that which requires hospitalization or causes death. Flu vaccine efficacy does not encompass prevention of infections. Third, the efficacy of flu vaccination is lowest among people who are 65 years old and older, particularly those with existing chronic conditions. Thus, in the Introduction, it is not accurate to state that this group is the “ideal target for vaccines” (rather, they are a high priority target) or to state that “for influenza vaccination to be considered effective, uptake in high-risk groups should approach 75%” (rather, to achieve the greatest benefit from influenza vaccination, uptake in high-risk groups should approach 75%.”) The overall effectiveness/efficacy of influenza vaccination is primarily determined by how the flu virus strains in the vaccine match the predominantly circulating strains.
- On page 2, line 67, it is stated that “the exact reasons for low vaccine uptake are unclear.” This is not accurate. There has been much research, including extensive literature reviews and meta-analyses, examining low influenza vaccine uptake across all populations, particularly those 65 and older and/or those with chronic health conditions. I would highly recommend using 1) Schmid P, Rauber D, Betsch C, Lidolt G, Denker M-L (2017) Barriers of Influenza Vaccination Intention and Behavior – A Systematic Review of Influenza Vaccine Hesitancy, 2005 – 2016. PLoS ONE 12(1): e0170550. doi:10.1371/ journal.pone.0170550 and 2) Nowak, G. J.; Sheedy, K.; Bursey, K.; Smith, T. M.; Basket, M. Promoting influenza vaccination: insights from a qualitative meta-analysis of 14 years of influenza-related communications research by U.S. Centers for Disease Control and Prevention. 2015 33(24):2741-2756. Language: English. DOI: 10.1016/j.vaccine.2015.04.064.
- Lines 135-140 – it would be helpful if an explanation were provided for why so many respondents did not answer all the questions included in the analysis. It is a very large drop from a total sample of 16,480 (i.e., 5051+5826+5603) to 3,335 respondents in the final analyses. Why such a large drop? Also, what, if anything is known or suspected about how those who completed the survey differ from those who failed to answer all the questions. Finally, this is a something that needs to be included in the Limitations (i.e., the large number failing to complete the survey and possible differences).
- It is unclear why respondents who had received their last influenza vaccination more than 1 year ago were grouped with respondents who had never received the influenza vaccine. These are two distinct groups. The first represents people who have direct experience with influenza vaccination, while the latter are “non-doers.” This study would be significantly stronger if the analyses had compared three groups: 1) those who recently received an influenza vaccination, 2) those who had received an influenza vaccination in the past, and 3) those who had never received an influenza vaccination.
- In the Results and Discussion, the phrasing should primarily indicate that associations or correlations were found rather than “the most essential factors that contributed to influenza vaccination” or “protective factors.” Higher educational level, for instance, was correlated with higher influenza vaccination, it was not a protective factor. Also, it should be noted in the Discussion and Conclusion that very few of the associations (or differences) were great. That is, the statistically significant influenza vaccination uptake differences that were found usually involved differences of 3-5%.
- Another major limitation is the reliance on socio-demographic and satisfaction measures in the surveys. Without having any items related to influenza disease and influenza vaccine risk-benefit beliefs and perceptions, this study cannot provide insights into why influenza vaccine uptake is quite low and decreasing in Hungary.
Reviewer 5 Report
Thank you for having the opportunity of reviewing your manuscript.
The main objective of this study was to identify the factors that influence influenza vaccination using questionnaire responses provided by a nationwide representative sample in Hungary.
Major comments
1. Please compare the characteristics of the responder and non-responder in Table 1.
2. There are a lot of Tables in the manuscript. Please focus on the aim of this study.